



# How much does traffic contribute to benzene and PAH air pollution? Results from a high-resolution North American air quality model centered on Toronto, Canada

Cynthia H. Whaley[1,2], Elisabeth Galarneau[1], Paul A. Makar[1], Michael D. Moran[1], and Junhua Zhang[1]

[1]Air Quality Research Division, Environment and Climate Change Canada, Toronto, Ontario, Canada.
[2]Climate Research Division, Environment and Climate Change Canada, Victoria, British Columbia, Canada.
**Correspondence:** Elisabeth Galarneau (elisabeth.galarneau@canada.ca)

**Abstract.** Benzene and polycyclic aromatic hydrocarbons (PAHs) are toxic air pollutants that have long been associated with motor vehicle emissions, though the importance of such emissions has never been quantified over an extended domain using a chemical transport model. Herein we present the first application of such a model (GEM-MACH-PAH) to examine the contribution of motor vehicles to benzene and PAHs in ambient air. We have applied the model over a region that is centered on Toronto, Canada, and includes much of southern Ontario and the northeastern United States. The resolution (2.5 km) was the highest ever employed by a model for these compounds in North America, and the model domain was the largest at this resolution in the world to date. Using paired model simulations that were run with vehicle emissions turned on and off (while all other emissions were left on), we estimated the absolute and relative contributions of motor vehicles to ambient pollutant concentrations. Our results provide estimates of motor vehicle contributions that are realistic as a result of the inclusion of atmospheric processing, whereas assessing changes in benzene and PAH emissions alone would neglect effects caused by shifts in atmospheric oxidation and particle/gas partitioning. A secondary benefit of our scenario approach is in its utility in representing a fleet of zero emission vehicles (ZEV), whose adoption is being encouraged in a variety of jurisdictions. Our simulations predicted domain-average on-road vehicle contributions to benzene and PAH concentrations of 4-21% and 14-24% in the spring-summer and fall-winter periods, respectively, depending on the aromatic compound. Contributions to PAH concentrations up to 50% were predicted for the Greater Toronto Area, with a domain maximum of 91%. Such contributions are substantially higher than those reported in national emissions inventories, and they also differ from inventory estimates at the sub-national scale because those do not account for the physico-chemical processing that alters pollutant concentrations in the atmosphere. The removal of on-road vehicle emissions generally led to decreases in benzene and PAH concentrations during both periods that were studied, though atmospheric processing (such as chemical reactions and changes to gas/particle partitioning) contributed to non-linear behaviour at some locations or times of year. Such results demonstrate the added value associated with regional air quality modelling relative to examinations of emissions inventories alone. We also found that removing on-road vehicle emissions reduced spring-summertime surface $O_3$ volume mixing ratios and fall-wintertime $PM_{10}$ concentrations each by ~10% in the model domain, providing further air quality benefits. Toxic equivalents contributed by vehicle emissions of PAHs were found to be substantial (20-60% depending on location), and this finding is particularly



relevant to the study of public health in the urban areas of our model area where human population, ambient concentrations, and traffic volumes tend to be high.

## 1  Introduction

It is well known that vehicle traffic emissions cause significant air quality (e.g., WHO (2005); Han and Naeher (2006);
Zhang and Batterman (2013); Farrell et al. (2016); Zimmerman et al. (2016); Gentner et al. (2017); Wang et al. (2017)) and greenhouse gas pollution (e.g., Sims et al. (2014); Zimmerman et al. (2016); Boulton (2016); US EPA (2002)) globally. In North America, vehicle emission controls have gradually reduced emissions of many pollutants, and made vehicles more fuel-efficient (U.S. Environmental Protection Agency, 2012; Reid and Aherne, 2016), with transportation policies being closely aligned in Canada and the United States. Governments and automobile manufacturers have both pledged to further reduce
emissions over the coming decades with increased adoption of fully-electric and hybrid-electric vehicles and charging infrastructure (Government of Canada, 2018). Additionally, the phase-out of high emission electricity generation has already begun, with Ontario's power generation 90% emissions-free as of 2015 (58% nuclear, 23% hydro, 9% wind and solar; Ontario (2019)). With this rapidly approaching future in mind, atmospheric chemistry models are useful tools for predicting the expected changes in pollutant concentrations that will result from a continuing reduction in vehicle emissions.

Of particular interest are highly toxic pollutants such as benzene and polycyclic aromatic hydrocarbons (PAHs), which are ubiquitous in the environment and include compounds that are carcinogenic, mutagenic, and teratogenic. In Canada, both have been subject to risk management under the Canadian Environmental Protection Act (CEPA) with actions focused on emergency management, fuel composition, and emitting activities associated with the natural gas, aluminum, iron and steel, and wood preservation industries (ECCC, 2015, 2018b). Ontario, Canada's most populous province and home to the nation's
largest city, Toronto, has developed health-based ambient air quality criteria for these pollutants, but these are exceeded at many locations throughout the country (Galarneau et al., 2016) despite the actions taken under CEPA. In the U.S., benzene and PAHs have been identified as contributors to excess cancer risk under the National Air Toxics Assessment (NATA) program (EPA, 2015).

National benzene emissions in Canada are compiled through the National Pollutant Release Inventory (NPRI) (NPRI, 2016)
but only for major industrial, commercial, and institutional sources. Previous model-based estimates (Stroud et al., 2016) suggest that 40-54% of benzene in the ambient air of major Canadian cities is due to mobile sources (e.g., cars, trains, ships), which are not included in the NPRI. In the U.S., the most recent National Emissions Inventory (NEI) (US EPA, 2018) includes benzene emissions estimates from a variety of natural and anthropogenic sources. At the national scale, 47% (90 kT) of benzene emissions in the U.S. are estimated to arise from mobile sources, of which 60% (54 kT) is from on-road vehicles (e.g.,





cars, trucks, motorcycles). Those on-road vehicle contributions range from 0-84% of total benzene emissions when reported at the county or tribal level.

Canadian emissions of four PAHs (benzo[b]fluoranthene, benzo[k]fluoranthene, benzo[a]pyrene, and indeno[1,2,3-cd]pyrene) from all known anthropogenic sources are estimated through the comprehensive national Air Pollutant Emission Inventory (APEI) (ECCC, 2018a) whose major point source emissions are reported through the NPRI. Mobile source contributions in

the APEI accounted for 8.3% (2,620 kg) of the total anthropogenic emissions of benzo[a]pyrene (31,516 kg) in 2017, the most recent data year available, consistent with (Environment Canada and Health Canada, 1994; Galarneau et al., 2007). In the U.S., the 2014 NEI (US EPA, 2018) reports that on-road vehicle emissions are 20% (28,931 kg) of total national anthropogenic benzo[a]pyrene (BaP) emissions (145,102 kg). Relative mobile source contributions are expected to be greater in urban centres (Nielsen, 1996; Harrison et al., 1996; Dunbar et al., 2001; Shen et al., 2011; Pachón et al., 2013; Kuoppamäki et al., 2014;

Miao et al., 2015) than they are at the national scale due to the spatial concentration of urban on-road vehicle use and the tendency of large industrial sources to be located outside those cities.

Here, we make use of a recently developed and validated high-resolution, on-line chemical transport model (GEM-MACH-PAH, Whaley et al. (2018b)) to study the impact of on-road vehicle emissions on ambient concentrations of benzene and a suite of PAHs in a regional domain centred over Toronto, Canada that includes much of sourthern Ontario and the northeastern

U.S. (Figure 1). GEM-MACH-PAH was run with identical meteorology for two emissions cases: (1) a "base" case with all emissions of all species and sectors included; and (2) a "no mobile" case with emissions of all species from on-road vehicles set to zero (BENZ, PAHs, and criteria air contaminants (CACs) such as $NO_x$, CO, VOCs, PM, etc). The vehicle contributions are determined from the difference of the "base" and "no mobile" cases, and this strategy has permitted us to calculate vehicle contributions in a realistic way that incorporates not only the effect of benzene and PAH emissions, but also the effect of

atmospheric processing caused by the changes to CACs emitted by motor vehicles. The "no mobile" scenario has additionally allowed us to quantify the impact of a hypothetical future fleet of zero-emissions vehicles (ZEV), whose adoption is being encouraged in a variety of jurisdictions. We did not simulate biofuel emission scenarios, as those fuels have sometimes been shown to increase PAH emissions, rather than reduce them (Karavalakis et al., 2011), and further work is needed before they can be simulated with confidence.

These simulations provide consistent information about the spatial distribution of concentrations and on-road vehicle contributions for benzene and PAHs. While other PAH chemical transport models exist (Aulinger et al., 2007; Friedman and Selin, 2012; San José et al., 2013; Gariazzo et al., 2014, 2015; Thackray et al., 2015; Zhang et al., 2016, 2017), this is the first study to use such a model to evaluate traffic contributions to ambient air and assess the change in resulting airborne toxicity. Our simulations also have the highest resolution employed to date in a North American domain, and the largest high-resolution

domain compared to other PAH modelling studies anywhere in the world.





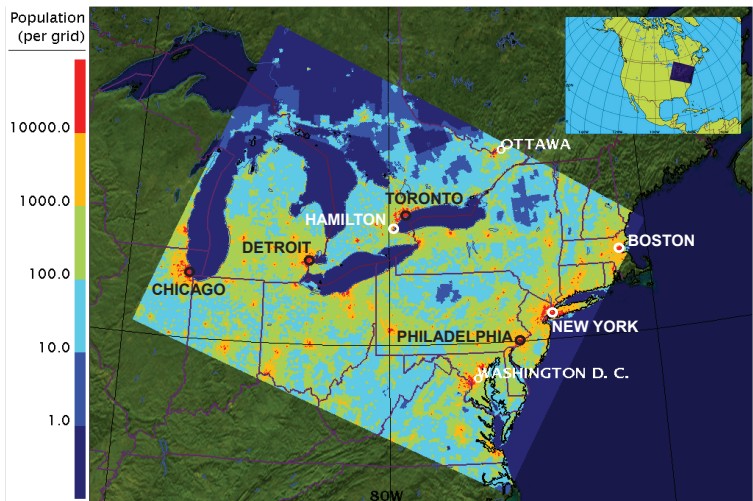

**Figure 1.** Model domain, coloured on a logarithmic scale by the human population per $2.5 \times 2.5$-km$^2$ model grid cell.

## 2 Methods

### 2.1 Model description

GEM-MACH (Moran et al., 2010) (Global Environment Multiscale Modelling Air quality and CHemistry) is an online chemical transport model driven by meteorological fields produced by the GEM numerical weather prediction model (Côté et al., 1998b, a). The model was recently adapted to include the emission, advection and diffusion, deposition, and chemical degradation of benzene (BENZ) and seven PAHs: phenanthrene (PHEN); anthracene (ANTH); fluoranthene (FLRT); pyrene (PYR); benz[a]anthracene (BaA); chrysene (CHRY); and benzo[a]pyrene (BaP) (Whaley et al., 2018b). While BENZ in ambient air is gaseous, PAHs are semi-volatile species, found in both the gas and particle phases. Their particle/gas partitioning in GEM-MACH-PAH is determined via the Dachs-Eisenreich scheme (Dachs and Eisenreich, 2000; Whaley et al., 2018b). Section I of the Supplement provides further information on PAH process representations within the model.

GEM-MACH-PAH was run at 2.5 km horizontal grid spacing on a domain that includes large North American urban areas such as Toronto, New York City, Chicago, Washington, D.C., Philadelphia, Boston, and Detroit (Figure 1).

Two time periods in 2009 were chosen (spring-summer, 13 May to 13 August; and fall-winter, 23 October to 5 January 2010) to balance the computational demands required for this model (IBM Power7 supercomputer) against the ability to examine seasonal differences and include evaluation data from a temporally coincident, high-density campaign conducted in 2009 west of Toronto (Anastasopoulos et al., 2012). Additional details about the model setup and run strategy are provided in the Supplement (Section I).



## 2.2 Emissions

Hourly, gridded, and speciated input emissions fields were prepared using the Sparse Matrix Operator Kernel Emissions sys-
tem (SMOKE, https://www.cmascenter.org/smoke, (Houyoux et al., 2002)) making use of criteria air pollutant emissions from
Canada's 2010 APEI (Sassi et al., 2015) reported by province, and U.S. EPA 2011 NEI (Eyth et al., 2013) emissions reported
at the county/tribal level. BENZ and PAHs were speciated relative to aggregate VOC emissions using VOC speciation pro-
files from the Canadian Emissions Processing System (Moran et al., 1997) for BENZ and special speciation profiles developed
by Galarneau et al. (2007, 2014); Whaley et al. (2018b) for PAHs. PAH emissions from on-road mobile sources were calculated
from VOC emissions generated using MOBILE 6.2C (EPA, 2002) and MOVES 2010b (https://www.epa.gov/moves/moves2014-
and-moves2010b-versions-limited-current-use) for Canada and the U.S., respectively. PAH species emissions were estimated
using PAH-to-VOC and PAH-to-organic carbon emission ratios from MOVES2014 converted to a total organic gas (TOG)
basis. Note that reported emission factors (EFs) for PAHs in the literature are highly variable. Different EFs were tested in the
model but those from MOVES2014 achieved the best results compared to observations (Whaley et al., 2018b).

SMOKE uses spatial surrogate fields to distribute vehicle emissions reported for each jurisdiction (e.g., provinces in Canada,
counties in the U.S.) among model grid cells. Unlike the MOBILE 6.2C-based inventory for Canada, the MOVES2010b-based
inventory for the U.S. explicitly includes an "off-network" road type that accounts for emissions when vehicles are stationary
(e.g., idle, parked, starting or refuelling), and this road type contributes ∼60% of on-road emissions (NEI2011). The effect of
this spatial-allocation difference between the MOBILE and MOVES inventories on the modelled on-road vehicle contributions
of BENZ and PAH is presented later in this work.

To construct the emissions fields for the"no mobile" case, emissions from all area sources, off-road mobile sources, and
minor point sources present in the "base" case were retained, but all on-road vehicle emissions (of all species) were removed.

## 2.3 Model evaluation summary

Detailed model descriptions and evaluations of GEM-MACH appear in (Moran et al., 2010, 2013; Makar et al., 2015b, a;
Gong et al., 2015; Whaley et al., 2018a) for pollutants other than benzene and PAHs. The GEM-MACH-PAH base simulation
used for the current study was previously evaluated in the most rigorous comparison to measurements yet published for such
a model at fine spatial resolution (Whaley et al., 2018b). That evaluation compared GEM-MACH-PAH output to benzene and
PAH measurements from 121 and 35 network sites, respectively, from Canada's National Air Pollution Surveillance program
(NAPS), the U.S. National Air Toxics Trends Stations (NATTS), and the Canada-U.S. Integrated Atmospheric Deposition
Network (IADN), which all record 24-hour integrated air concentrations every one in six days, at locations associated with
a variety of population densities and land uses (e.g., urban, sub-urban, industrial and rural locations). Additional 2-week
integrated PAH measurements from 46 sites in a high-spatial-density campaign conducted in Hamilton, Ontario, Canada in
spring-summer and fall-winter 2009 (Anastasopoulos et al., 2012) were also used to assess concentration variability within a
city as well as within model grid squares (Supplement, Section I).



Ratios of modelled to measured concentrations were close to unity for the lowest molecular weight compounds in spring-summer, and increased modestly in the fall-winter and with increasing molecular weight. Modelled concentrations were found to be statistically unbiased relative to measurements (paired t-test with t < 1, p > 0.01) for all compounds and seasons except for BaP in fall-winter (Whaley et al., 2018b), which was biased high. Model output for the latter compound and season combination was therefore excluded from this study.

GEM-MACH-PAH's particle/gas partitioning parametrization was evaluated at six IADN stations and the results showed a substantial improvement over the previous AURAMS-PAH partitioning (Galarneau et al., 2014) due to an empirically-based update in partitioning parameters (Whaley et al., 2018b).

The good overall performance (Whaley et al., 2018b) of GEM-MACH-PAH demonstrates its validity for calculating ambient concentrations and for assessing source contributions at its evaluated resolution (2.5 km grid size and seasonal time scale).

**3  Results**

Spatial distributions of modelled concentrations and vehicle contributions were similar for BENZ and the seven PAHs. As a result, we focus on a few representative species in this section, and show results for the remaining species in the Supplement.

**3.1  Benzene and PAH concentrations from the base case**

Modelled "base" case (all emissions activated) average airborne concentrations of BENZ, PHEN, PYR and BaP are shown in
Figure 2 for the spring-summer and Figure S.2(a) for the fall-winter. The three PAH compounds exhibit a range of volatilities. PHEN and BaP are found predominantly in the gas and particle phases, respectively, whereas PYR, with a mid-range volatility, is typically found in both. The spatial distribution of concentrations is similar to the distribution of human population shown in Figure 1 as expected from the prevalence of anthropogenic sources in the study area.

Modelled concentrations for BENZ and PAHs are higher in fall-winter than in spring-summer (Whaley et al., 2018b) due to
lower fall-winter temperatures and solar radiation. These factors lead to to reduced photochemical degradation and increased vertical stability, which in turn induce less vertical mixing and dilution, and lower boundary layer heights. Thus, ambient concentrations are higher per unit emission in fall-winter than in spring-summer. Additionally, total emissions for PAHs are higher in fall-winter than in spring-summer (Figure S.3a) due to increased on-road vehicle emissions (e.g., cold starts) and combined area/off-road mobile sources (e.g., heating, snowmobiling). Note that different rates of PAH oxidation in the different seasons
are expected to lead to different rates of production of secondary products such as oxy- and nitro-PAHs. These secondary products are not yet included in GEM-MACH-PAH due to uncertainties in their sources and properties, but they are under consideration for future addition to the modelling package given that some of these compounds are more toxic than their parent PAHs.



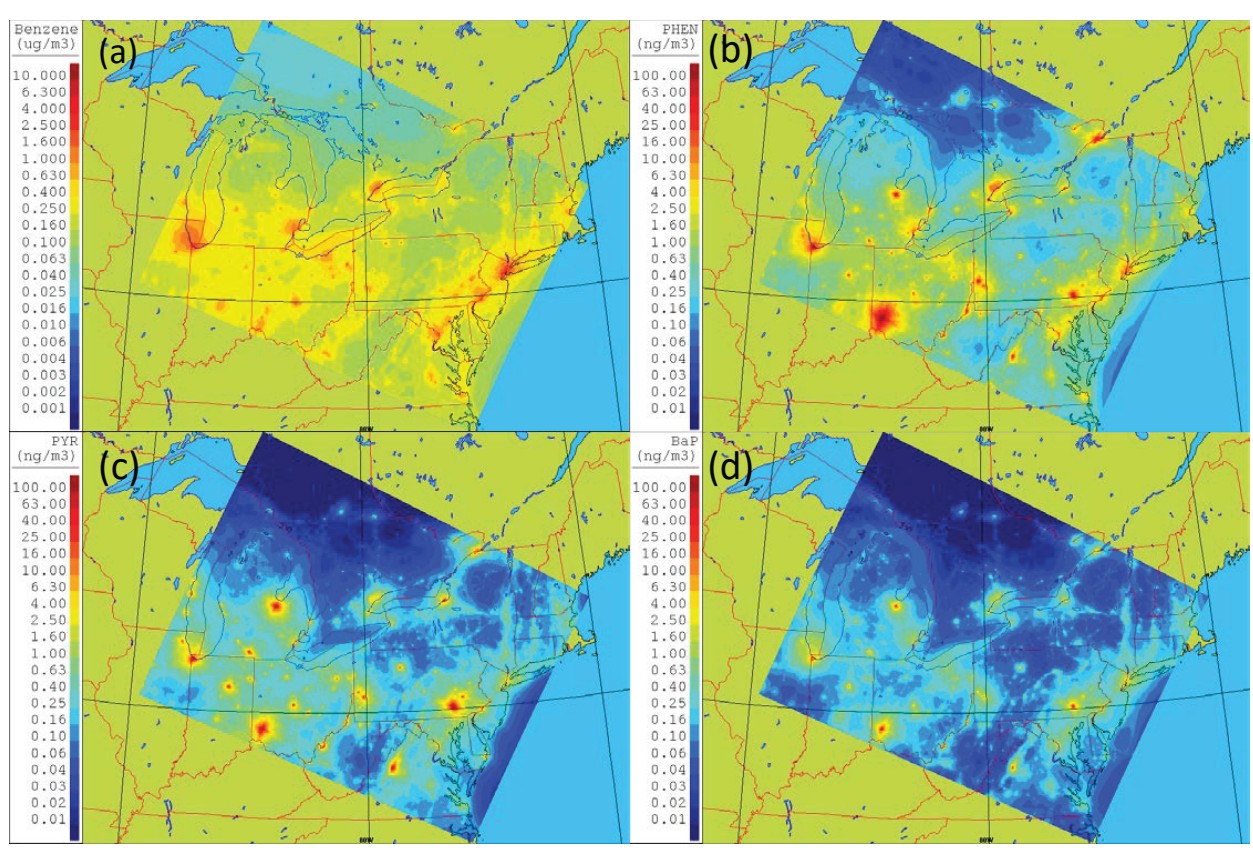

**Figure 2.** Modelled average airborne concentrations for (a) BENZ, (b) PHEN, (c) PYR, and (d) BaP in the spring-summer.

## 3.2 Absolute on-road vehicle contributions

Figure 3 and Figure S.2(b) shows the contribution of on-road vehicles to ambient concentrations in the spring-summer and fall-winter, respectively, as represented by the absolute differences in concentrations between the "base" and "no mobile" cases. Concentrations in the "no mobile" case are significantly lower than those for the "base" case as expected from the lack of on-road vehicle emissions.

Major cities are prominent in Figure 3 for all species as expected given high urban traffic volumes. In the Greater Toronto
Area (GTA), BENZ concentrations due to on-road vehicles in spring-summer are on the order of 0.1-0.3 $\mu$g m$^{-3}$ (Figure S.4) and these values are similar to those of other urban centres in Ontario such as Hamilton. Spring-summer contributions up to 0.5-0.9 $\mu$g m$^{-3}$ of BENZ are seen in the large urban centres of the U.S. such as New York City, Chicago, and Washington, DC (Figure S.4) as well as in several smaller U.S. cities (Figure 3).

Spatial distributions of absolute on-road mobile source contributions for the PAHs are similar to those for BENZ. Spring-
summer contributions in the GTA for PHEN, PYR and BaP are approximately 2.0, 0.35, and 0.3 ng m$^{-3}$, respectively, and





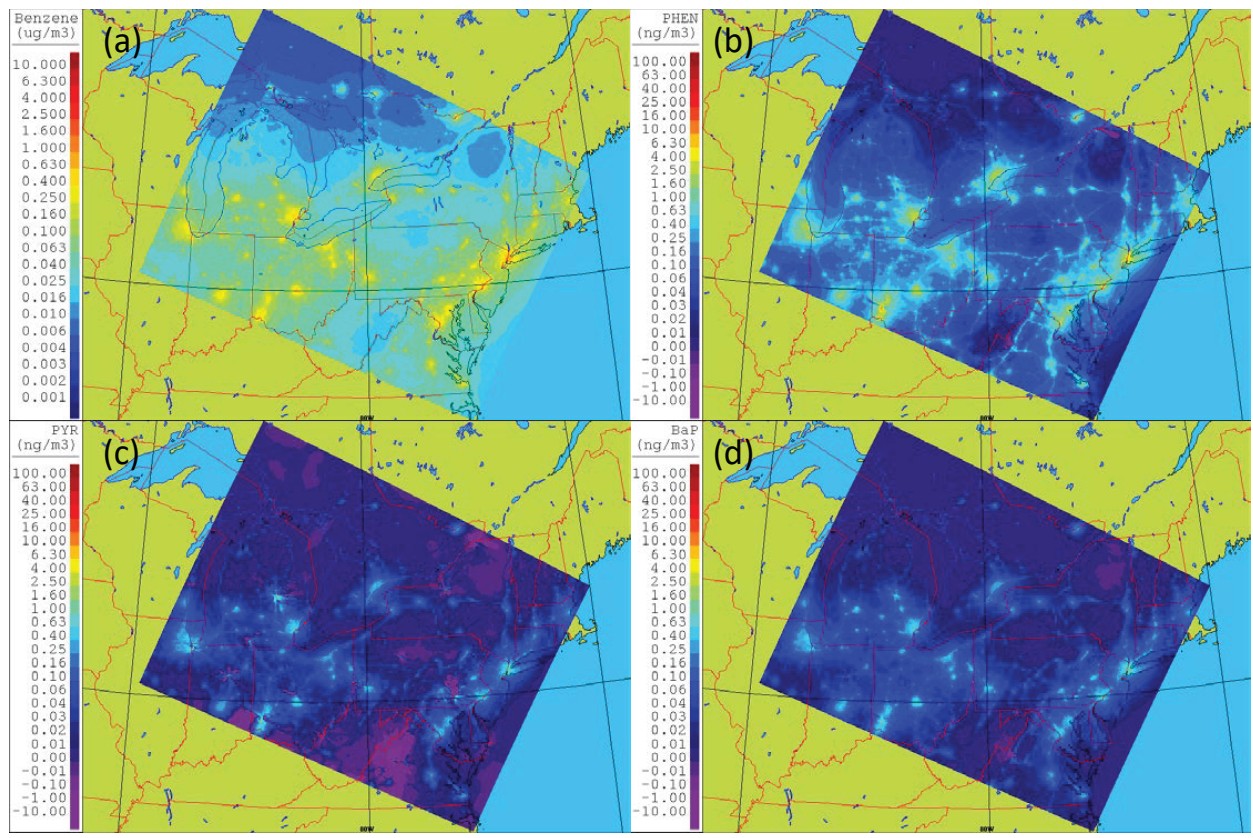

**Figure 3.** Seasonal-average absolute on-road vehicle contributions for (a) BENZ, (b) PHEN, (c) PYR, and (d) BaP in the spring-summer.

slightly higher in fall- winter for the species reported in Figure S.2b. Absolute contributions from on-road vehicles are higher in the major U.S. urban areas than they are in the Canadian cities (examples for BENZ and PYR shown in Figure S.4). These cross-border differences arise in part because of differences in the spatial surrogates mentioned above and the different emissions inventories (see Section 2.2 and Supplement, Section IV). Furthermore, the cities in the U.S. portion of the study
region have populations that are larger on average than the cities in the Canadian portion, and concentrations of PAHs have been shown to increase in direct proportion to human population (Hafner et al., 2005).

However, there are some geographically-limited exceptions where the removal of on-road vehicle emissions causes PAH concentrations increase slightly (areas with negative values in Figure 3): in the spring-summer there are small concentration increases in the northeastern portion of New York State and along the border between Virginia and West Virginia. These
apparent anomalies are located where base PAH concentrations are already relatively low, and are consistent with the impacts on oxidant chemistry discussed below, and in the supplement (Section V). In fall-winter, PAH increases in the "no mobile"



**Table 1.** Domain-wide average and maxima on-road vehicle contribution to ambient concentrations. NR = not reported

|  | benzene | PHEN | ANTH | FLRT | PYR | BaA | CHRY | BaP |
|---|---|---|---|---|---|---|---|---|
| spring-summer avg | 21% | 21% | 19% | 4% | 8% | 16% | 13% | 19% |
| spring-summer max | 74% | 91% | 86% | 64% | 76% | 75% | 74% | 83% |
| fall-winter avg | NR | 24% | 24% | 14% | 18% | 19% | 19% | NR |
| fall-winter max | NR | 72% | 64% | 52% | 64% | 49% | 54% | NR |

case are confined to two small regions near the domain borders (Figure S.2b), where factors other than the emission change may be responsible (e.g., boundary effects, numerical issues, etc.).

Nevertheless, measurements show that Ontario's annual ambient air quality criteria for BENZ (0.45 $\mu$g m$^{-3}$) and BaP (0.01 ng m$^{-3}$), the latter of which is used by the province as a surrogate for PAHs, are exceeded in Toronto, Hamilton, and Windsor (Galarneau et al., 2016). The absolute contributions of on-road vehicles in those areas (Figure 3) suggest that reducing their emissions could assist in reducing those exceedances. Policies and programs seek to achieve air quality benefits with minimal socioeconomic cost, thus knowledge of the relative (e.g., percent) contributions of different sources is an important criterion for prioritizing possible management actions. The reduction of on-road vehicle emissions will only be effective in achieving meaningful reductions in ambient concentrations if their local contributions are significant relative to the total.

### 3.3 Relative on-road vehicle contributions

The relative contributions (expressed as the percentage of the "base" case concentrations) of on-road vehicles to BENZ, PHEN, PYR, and BaP concentrations are shown as maps and frequency distributions of domain-wide ranges in Figures 4 and 5, respectively, with maps for the remaining PAH species shown in Figures S.2c and S.8 of the Supplement. Domain-wide average and maximum values are also listed in Table 1 and shown in Figure 5. Relative on-road vehicle contributions to PAH concentrations in individual model grid squares had maxima as high as 64-91% in spring-summer. Maxima were slightly lower in fall-winter (49-72%) for the subset of PAHs reported for that period (Figures 5 and S.2c, and Table 1). Domain means, however, were higher in the fall-winter than in the spring-summer (Figure 5, and Table 1). The highest relative on-road vehicle contributions were observed in or near small cities such as North Bay, Ontario; Columbus and Toledo, Ohio; and Grand Rapids, Michigan, where major highways are found in areas of otherwise low ambient concentrations.

In the spring-summertime, domain-mean on-road vehicle contributions to ambient BENZ in the GTA were on the order of 14-37%, consistent with, though slightly lower than previously reported values (Stroud et al., 2016) due to the latter study including off-road mobile sources in their "mobile" category. PAH contributions in the GTA ranged from 5-50% depending on species, season, and proximity to major highways. Even greater contributions were seen in other Canadian cities, thus, our results suggest that fewer and/or less extreme exceedances of provincial BENZ and PAH guidelines could be achieved by reductions in on-road vehicle emissions in cities within the model domain.

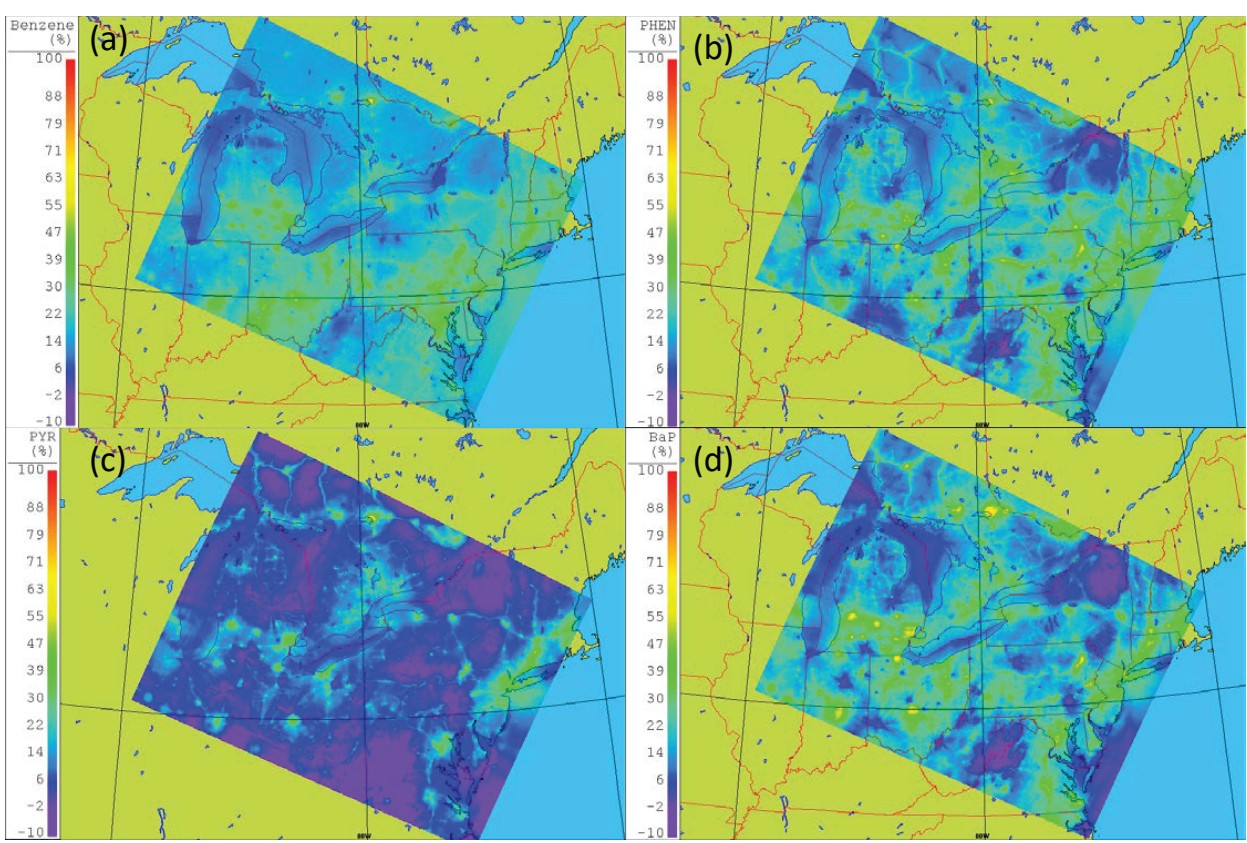

**Figure 4.** Seasonal-average relative on-road vehicle contributions to ambient concentrations of (a) BENZ, (b) PHEN, (c) PYR, and (d) BaP in the spring-summer.

This finding is significant in the Canadian policy-making context and demonstrates the value of examining pollutant emissions and concentrations on fine geographic scales. The APEI and other Canadian efforts (Environment Canada and Health Canada, 1994; Galarneau et al., 2007) have estimated that PAH contributions from on-road mobile sources amount to less than 10% of total anthropogenic PAH emissions, at the national scale (provincial-scale estimates for BENZ and PAHs are not included in the APEI). Such minor relative contribution at the national scale could lead to the neglect of the on-road mobile source category in emissions reductions strategies, yet we have shown that this category is important at the local scale in terms of impacts of potential BENZ or PAH management actions.

In the U.S., NEI emissions are reported at the county or tribal level. On-road vehicle contributions in those reported emissions are closer to this study's high-resolution results in ambient air than are the contributions in emissions reported at the national scale. Nonetheless, on-road vehicle contributions of BENZ and PAHs differ between emissions and ambient air due to physico-





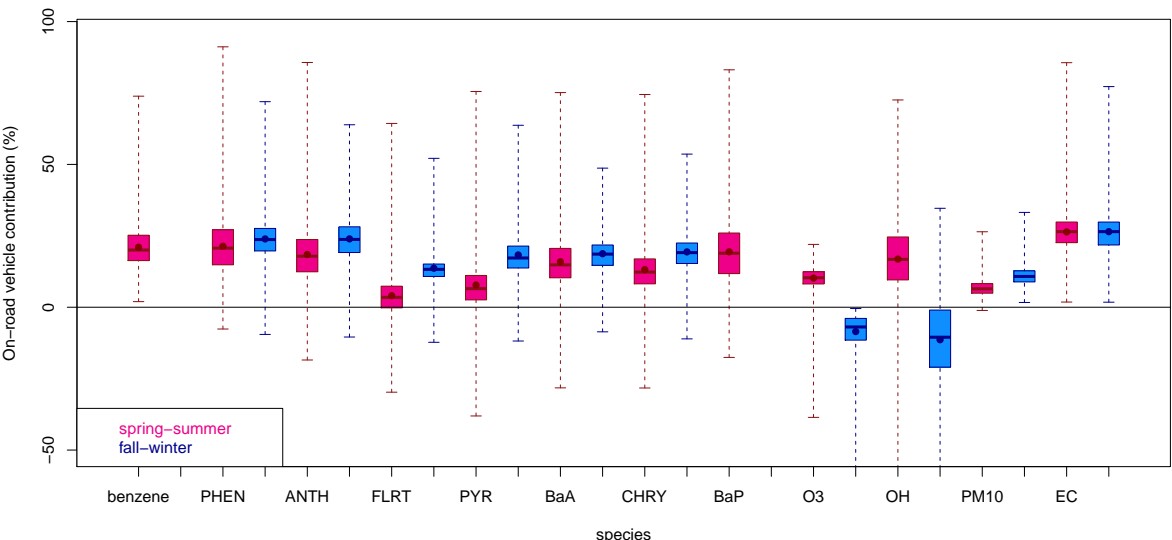

**Figure 5.** Seasonal-averaged relative on-road vehicle contributions to daily-average surface concentrations (in % of total) in the model domain for all pollutants studied. Whiskers extend to the maximum range of the data in the domain, the centre line is the domain-median, and the dots are the domain-average.

chemical processing that occurs in the atmosphere. Such processing varies with time of year and levels of vehicle co-pollutants as described below.

### 3.4 Seasons' impact on on-road vehicle contributions to benzene and PAHs

Though PAH emissions from on-road vehicles are higher in fall-winter than in spring-summer (Figure S.3a), the relative contribution of on-road vehicles to total emissions is stable among seasons (viz., domain-average differences between fall-winter and spring-summer emission contributions from on-road vehicles are -0.6% and range from -1.8% to +2.7% for species reported for both time periods; see Figure S.3b).

     In contrast, relative on-road vehicle contributions to ambient concentrations differ more between seasons than do their
emissions, with differences of +3% to +10%, respectively between fall-winter and spring-summer (Figure 5). This suggests that a given relative reduction in on-road vehicle emissions may lead to greater concentration reductions in fall-winter than in spring-summer, and highlights the importance of conducting analyses that represent conditions at different times of year. The following analysis expands on factors that are potentially responsible for this temporal variability.





### 3.5 Oxidants' impact on on-road vehicle contributions to benzene and PAHs

GEM-MACH-PAH includes reactions of BENZ and PAHs with two oxidants: hydroxyl radical (OH, which reacts with BENZ and gaseous PAHs) and ozone ($O_3$, which reacts with particulate BaP) (Galarneau et al., 2014; Whaley et al., 2018b). As noted earlier, the "no mobile" case zeroed on-road vehicle emissions for all emitted chemical species, including precursors to tropospheric OH and $O_3$, such as $NO_x$, CO, and VOCs. Thus, the removal of vehicle emissions impacts not only the concentrations of the pollutants of BENZ and PAH directly, but also modifies the concentrations of the oxidants responsible
for their chemical degradation.

In the spring-summer, the removal of on-road vehicle emissions of criteria air pollutants leads to OH reductions in most parts of the study region (Figure 5 and red areas in Figure S.9a). Oxidative removal rates of BENZ and gaseous PAH are thus reduced in those parts of the domain as a result. This contributes to the finding that reductions of BENZ and PAH emissions of 10-27% associated with the removal of the on-road mobile emissions (Figure S.3b) result in domain-average concentration reduction
to a lesser degree, of 4-21% (Table 1). The removal of all mobile on-road emissions decreases oxidant concentrations, hence BENZ and PAH from other sources is oxidized to a lesser degree, offsetting the reductions in BENZ and PAH associated with the mobile emissions removal itself. Conversely, OH increases by 10-50% in some urban cores (e.g., Toronto, Detroit, New York City; blue areas in Figure S.9a) in the spring-summer when on-road vehicle emissions are removed, in response to higher $O_3$ levels via reduced $NO_x$ titration, with a similar result over large portions of the study area in fall-winter (blue in Figure
S.9b). At these times and locations, a positive feedback is produced, whereby the degradation of BENZ and gaseous PAHs from other sources is accelerated in areas where their emissions from vehicles have been removed.

Similarly, on-road vehicles emissions in spring-summer contribute to a domain-wide median of ~10% (~3.5 ppbv) to surface $O_3$ volume mixing ratios (Figure 5). Thus, when vehicle emissions are removed, airborne BaP is reduced less than expected from the emissions reduction because of reduced oxidation of BaP from $O_3$. However, in and around major cities, the
changes in $O_3$ due to on-road vehicles are smaller than 10% and are often negative (see blue in Figure S.9c), viz., $O_3$ increases in response to the removal of $NO_x$ from on-road vehicles in this hydrocarbon-limited regime (Sillman, 1995; Kleinman et al., 2000; Sillman and West, 2009; Jing et al., 2014; Zhang et al., 2014), thus increasing oxidation of BaP from non-mobile sources. The "no mobile" case degradation of BaP is thus enhanced in urban areas. This leads to net urban BaP reductions that are greater than might be expected from the removal of urban BaP on-road vehicle emissions alone.

### 3.6 Effect of elemental carbon on on-road vehicle contributions to PAHs

PAHs are semivolatile and their mass is therefore partitioned between the gas and particle phases in ambient air. Particulate fraction (PF) (Junge, 1977), the ratio of the particulate concentration to the total (gaseous + particulate) concentration, is a common descriptor of particle/gas partitioning. Smaller, lighter PAHs have small PFs of about 0 (e.g., for PHEN), whereas larger, heavier PAHs, have PFs around 1 (e.g., for BaP), and semi-volatile PAHs like FLRT and BaA fall somewhere in the
middle (Figure 6a,c). The extent of PAH partitioning varies with temperature and with the availability and composition of particulate matter (PM), where the latter is affected by the zeroing of on-road vehicle emissions as discussed below.



On-road mobile source contributions to domain-averaged PM were 7% in spring-summer and 10% in fall-winter (Figure 5 and S.9e and f), lower than those for total (gaseous + particulate) PAHs. This suggests that PAH PFs might rise, that is, a greater relative amount of the PAH might partition to the particulate phase, if on-road mobile source emissions were reduced because

relatively more PM would be available per unit mass of remaining PAH. However, decreases in PAH PFs were observed in the "no mobile" case (Figure 6b, d-f). This was due to the nature of partitioning, which is specific to PM speciation. Elemental carbon (EC) is the prime sorbent for PAHs in the particle/gas partitioning parameterization in the model (Dachs and Eisenreich, 2000). The on-road vehicle contributions of EC averaged 26.5% of total EC mass in both seasons (Figures 5 and S.9g and h). Relative to the "base" case, EC in the "no mobile" case was thus reduced to a greater extent than PAH due to the high

EC fraction of PM emissions from motor vehicles. This in turn, resulted in the particle/gas partitioning equilibrium being shifted toward the gas phase since less EC mass was available to sorb the remaining PAH. Shifts in particle/gas equilibrium in turn affect removal processes such as deposition and degradation, whose mechanisms differ for gaseous and particulate compounds (Bidleman and Foreman, 1987). Further analysis of the differences in PAH lifetimes that arise from a shift in particle/gas partitioning is beyond the scope of this paper, but it should be kept in mind for future analyses, particularly those

that incorporate considerations of transboundary or long-range transport.

## 3.7 Sensitivity Considerations

The results described thus far have shown that on-road vehicle emissions contribute substantially to benzene and PAHs in ambient air at a variety of locations in our study area. Differences between seasonal vehicle contributions have been examined with respect to the atmospheric processing that transforms toxic pollutants after they have been emitted to the air. Potential

sensitivities of our model results to the uncertainties in emissions and atmospheric chemistry are explored in this section, and are described in more detail in the Supplement (Section V).

We expect that the largest contribution to uncertainty in our results to be associated with the PAH mobile emissions. The on-road vehicle EFs for PAHs that underlay this study's inventory were taken from MOVES2014b (EPA, 2014). These factors were determined from two U.S. reports that examined gasoline and diesel emissions separately (Kishan et al., 2008; Khalek et al.,

2009). We carried out four sensitivity simulations with GEM-MACH-PAH with the BENZ and PAH emissions from on-road vehicles scaled by factors of 0.5 and 2 in both seasons. This range corresponds approximately to the 25th and 75th percentiles of the range of EFs reported in the recent peer-reviewed literature (Whaley et al., 2018b). The model responded consistently to on-road vehicle emission scaling with average changes to the vehicle contribution amount of -5 to -10% and +20 to +30%, depending on species, for halved and doubled vehicle emissions, respectively (Table 1 vs 2). This finding suggests that the

relative importance of vehicle contributions when different EFs are used remains consistent with our current results across a broad range of emissions levels. This topic is described in more detail in the Supplement (Section V).

Whereas the effects of emissions perturbations are straightforward to evaluate, uncertainties that arise from atmospheric chemistry are complex to assess because they result from secondary formation processes for atmospheric oxidants. A reduction in precursor emissions yields a nonlinear change in oxidant concentrations that depends on chemistry and the physical

state of the atmosphere at each location. The removal of on-road vehicle emissions induces a large range of changes in oxidant





**Figure 6.** (a & c) Sample maps of particulate fraction (PF), and (b & d) its percent change due to the removal of on-road vehicle emissions. Spring-summer FLRT and BaA shown as examples. The spring-summer and fall-winter averages of absolute (e) and percent (f) reduction in PF for all PAH species.





**Table 2.** Domain-wide average on-road vehicle contributions to ambient concentrations, when on-road vehicle emissions of BENZ and PAHs are halved or doubled. NR = not reported

|  | benzene | PHEN | ANTH | FLRT | PYR | BaA | CHRY | BaP |
| --- | --- | --- | --- | --- | --- | --- | --- | --- |
| spring-summer test with 0.5× emissions | 11% | 8% | 9% | 0.5% | 2% | 7% | 6% | 6% |
| spring-summer test with 2× emissions | 42% | 30% | 33% | 4% | 10% | 27% | 21% | 25% |
| fall-winter test with 0.5× emissions | NR | 12% | 4% | 8% | 10% | 11% | 11% | NR |
| fall-winter test with 2× emissions | NR | 41% | 41% | 27% | 34% | 30% | 30% | NR |

concentrations (Figures 5 and S.9). The resulting changes in BENZ and PAH concentrations as a function of changes in oxidant concentration are highly variable (e.g., Figures S.5 and S.6), and some geographic areas see a net increase in PAH concentrations because reduced atmospheric oxidation of PAHs overwhelms the effect of removing vehicle emissions. However, such results were uncommon throughout the study area. For a reactive PAH such as pyrene, for example, 88.3% and 99.9% of model

grid squares in spring-summer and fall-winter, respectively, responded to the removal of vehicle emissions with reductions in ambient PAH concentrations. Relatively unreactive benzene, on the other hand, responded to emissions reductions with ambient concentration reductions in all model grid squares.

### 3.8   Human Health Implications

The removal of on-road vehicle emissions would lead not only to reductions in ambient benzene and PAH concentrations

(as well as in other pollutants), as demonstrated by our model results, but also to reductions in human exposure. Proximity to roadways and traffic has been linked to elevated exposure outdoors, and this has led to particular concerns for commuters (Miao et al., 2015; Yan et al., 2015; Tan et al., 2017; Lovett et al., 2018; Miri et al., 2018). Further inhalation exposure to traffic pollutants occurs in indoor environments, where infiltration of outdoor air can contribute a substantial proportion of benzene and PAH exposure (Naumova et al., 2002; Xu et al., 2016), adding to concerns about residences, schools and workplaces that

are situated near roadways.

PAH species vary in toxicity, and their mixture is often represented as a toxic equivalent concentration (TEQ) which is the sum of contributing compound concentrations that have been normalized by their carcinogenic potencies relative to BaP (Nisbet and LaGoy, 1992). The percent reduction in TEQ when on-road vehicle emissions were removed (Figure 7) averaged 18.9% across the domain. The magnitudes and geographic distribution of these TEQ reductions closely follow the reductions

in simulated PAH concentrations, implying a direct toxicity benefit of mobile emissions reductions. For large urban areas and their suburbs, where both ambient concentrations and human population density are high, results herein suggest that TEQs could be reduced by values of 20-60% if vehicle emissions were removed. Maximum TEQ reductions of up to ∼80% were predicted for some rural and suburban locations near highways (e.g., North Bay, ON; Sudbury, ON; Grand Rapids, MI; and Maumee, OH).

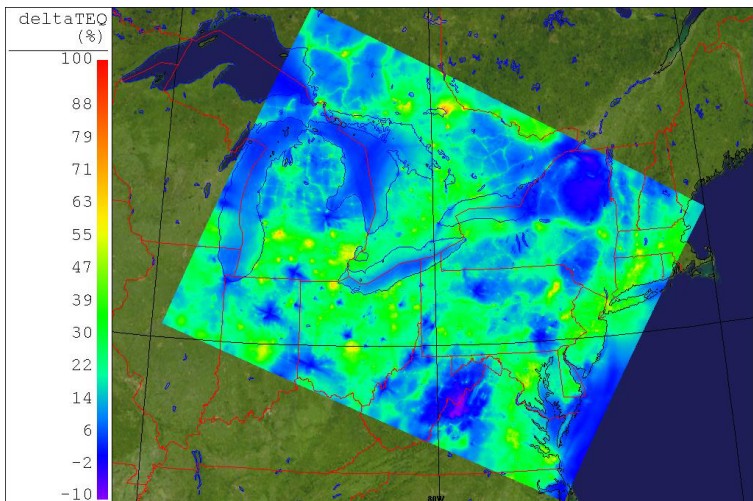

**Figure 7.** Average change in BaP toxic equivalents (TEQ) in ambient air during spring-summer 2009 when on-road vehicle emissions are set to zero.

Benzene has not been assigned a BaP toxic equivalency factor in the available literature. However, the combination of its modelled concentration and its human toxicity potential (Hertwich et al., 2001), which are approximately 1000 times larger and smaller than those of BaP, respectively, suggests that reductions in traffic emissions would lead to similar reductions in risk for benzene as for PAHs.

Further connection of the results of this study to potential human health benefits will require careful attention to the interplay
between air toxics and criteria air contaminants, since these are not often considered together in air quality research. The development, evaluation and first application of GEM-MACH-PAH makes such work possible. Another priority for future research is the improvement of emissions inventories and model process representations.

## 4    Conclusions

The previously validated GEM-MACH-PAH model was used to simulate benzene, PAH, and other pollutant concentrations for
both base case and no-mobile emissions scenarios for a densely populated region in north-eastern North America. Taking the difference of the two scenarios has allowed the on-road vehicle contribution to ambient concentrations to be calculated and this effect was 4-21% for benzene and PAHs, and 10% for both spring-summer $O_3$ and fall-winter $PM_{10}$ on average in our southern Ontario/north-eastern U.S. model domain (variations for season and compound). Maxima seasonally-averaged vehicle contributions were 74% for BENZ, 91% for PHEN, and 22% for spring-summer $O_3$ and 33% for fall-winter $PM_{10}$ within the
model domain. These can additionally be interpreted as the relative reductions in pollutant concentrations expected with the

introduction of a ZEV fleet. The chemical transport modelling of benzene and PAHs presented in this study is unprecedented in terms of combined domain size and spatial resolution, and it has demonstrated that vehicular sources of these toxic species make substantial contributions to ambient concentrations (expressed on the basis of both mass and toxic equivalents) at the urban scale. This suggests that meaningful decreases in BENZ and PAH concentrations can be achieved through on-road

vehicle emission reductions. Such reductions could be achieved through a number of potential management actions, including increases in ZEV use, and greater use of active transportation modes such as walking and cycling. Future work aims to include more PAH species, including secondary reaction products such as oxy- and nitro-PAHs, and to improve model representation of wintertime benzene and BaP.

*Code availability.* The MACH-PAH (chemistry) code is available here:

https://zenodo.org/record/1162252#.Wm9DtK1lJZQ, DOI:10.5281/zenodo.1162252, and the GEM (meteorology model) code is available here:
https://github.com/mfvalin?tab=repositories. The executable for GEM-MACH-PAH is obtained by providing the chemistry library (MACH-PAH) to GEM when generating its executable.

*Author contributions.* EG, PM, and CHW designed the model experiment and CHW developed the model code and simulation set-up. JZ,

MDM, and CHW created the emissions files for the model simulations. CHW performed the model simulations and analysis and CHW and EG wrote the manuscript with support from all co-authors.

*Competing interests.* We have no competing interests.

*Acknowledgements.* The authors acknowledge funding from the Government of Canada's Program of Energy Research and Development (PERD; ALMITEE Project led by Jeffrey R. Brook) and from Environment and Climate Change Canada's Climate Change and Air Pollution

program. The authors also acknowledge the contributions of Ayodeji Akingunola, Sylvie Gravel, Wanmin Gong, Craig Stroud, and Qiong Zheng for their assistance in setting up GEM-MACH-PAH (Whaley et al., 2018b). All maps were created using CMC software, SPI.



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
