# Peer review of "How much does traffic contribute to benzene and PAH air pollution? Results from a high-resolution North American air quality model centered on Toronto, Canada"

_Atmospheric Chemistry and Physics, 2019_

## Referee Comment (RC1) · Anonymous Referee #1 · 15 Nov 2019

General comments:

This paper summarizes a study in which the finely resolved (2.5x2.5 km) GEM-MACH-PAH model was employed to investigate contributions of motor vehicles to benzene and several PAHs in ambient air across northeastern North America. I found this paper to be a significant contribution to the field and of scientific merit, and had relatively few critical comments. I was particularly interested in the combined effects of removing benzene/PAHs at the same time as removing air quality criteria pollutants (precursors of compounds that degrade benzene/PAHs), and the differences in these effects in ru-

Interactive
comment

ral versus urban locations. The study addresses several interesting questions on a very fine spatial scale, including: -How much do vehicles contribute to ambient benzene and PAH concentrations, and how does that differ between the warm and cold seasons and across urban versus rural locations? -How do reductions in vehicle emissions play out in actual ambient concentrations? -What are the combined impacts of removing benzene/PAH oxidant precursors and benzene/PAH emissions themselves on ambient benzene/PAH concentrations? -How does particulate fractioning respond when all vehicular emissions are removed? -How do benzene/PAHs respond in general to changes in oxidant concentrations? -How sensitive is the model to doubling and halving vehicular emissions? -How does human health risk change when vehicular emissions are removed?

Specific comments:

Line 121: Can examples of "off-road mobile sources" be provided?

Line 135: It would be helpful to have "close to unity" qualified so readers don't have to dig back through the previous Whaley paper. Plus or minus what on average?

Line 155: Lower temperatures also increase partitioning to particles for PAHs. Does the decrease in degradation rates and decrease in dilution outweigh these effects?

Line 180-ish: It was mentioned in the intro that the adoption of ZEVs was encouraged in a variety of jurisdictions. Are there differences in this "encouragement" between US and Canadian cities that could also account for the higher contributions from on-road vehicles in US cities?

Figure S2. Is there a particular reason why the BENZ and BaP fall-winter plots are not shown? It would be handy to be able to compare seasonally for all four species chosen, or have an explanation as to why they are not shown. It is mentioned in the conclusions that future work aims to improve model representation of wintertime benzene and BaP, but (unless I missed it) I couldn't find a discussion about why their representation wasn't

acceptable.

Lines 189-192: Would be helpful to quantify, based on model results, how much lower concentrations would need to go to reach these criteria, and whether the complete removal of on-road vehicle sources reverses exceedances, or if additional sources would need to be removed (hard to tease out from the plots).

Line 235: This is probably discussed in previous papers, but would be handy to have a clarification here: BaP is the ONLY PAH to react with O3 on particles, even though partitioning to particles takes place for the other PAHs as well? Given the changes in O3 in rural and urban areas and the impact on BaP in the "no mobile" case, can the authors comment on how including O3 particle-phase oxidation for the other PAHs might impact their reductions? Would it be similar? How would the combination of changing OH and O3 concentrations play out for semi-volatile species? If it's easy to test this, I suggest adding it to the study.

Sensitivity to oxidant experiments shown in the Supporting Info: This is a very interesting section of the paper and it's too bad it can't be highlighted in the main text. I think it would be helpful if the authors put a section in the methods that describe how the sensitivity analyses were conducted for making the S5 and S6 plots.

Technical Corrections:

The placement of the parentheses in the very first line of the Introduction (lines 29-31) makes the sentence difficult to read. I think it would help to have another descriptor after "air quality", like "degradation", or "problems".

Line 124: Again, parentheses are oddly placed. I suspect automated citation management software was used, so suggest combing back through the paper and removing those interrupting parentheses.

Line 165: Should be "show" instead of "shows"

Line 183: Put a "to" between "concentrations" and "increase".

Line 339: Should it read "91% PAH" instead of "91% PHEN"?

---

## Referee Comment (RC2) · Anonymous Referee #2 · 20 Dec 2019

The introduction needs to be much clearer on why this reserach was undertaken. Additionally, the assertions need to be backed up with appropriate citations. In particular, Lines 33-35 do not seem true, so need citations to support them.

I cannot comment on how this model performs relative to its peers. However, I don't see the utility in turning various block-level emission estimates and county level emissions estimates into a geographical model. There does not appear to be any need of a 2.5km resolution for the type of analyses presented here.

[Figure]

In examining the referenced article Whaley et al 2018b, it is not clear that the GEM-MACH-PAH model has the accuracy and precision to really describe the difference the authors say it is between the two scenarios. The variability across sites and seasons seems that it is larger than the observed difference. For example, the % reduction of the PF of FLRT in winter looks to be 20% in Figure 6f. In Whaley et al 2018b, the model to measurement ratio for FLRT varies across sites from -10 to 10. It seems that this degree of uncertainty makes it hard to believe the model is able to tell the difference between a change of 20% and a change of 200%.

I'm also not completely clear on what the difference is between the Whaley 2018 model and the model used in this paper.

---

## Author Comment (AC1) · 27 Jan 2020

Response to reviewers: Reviewers comments are in italic font, authors' comments/responses start with (AC), and revised text is shown in red font

Anonymous Referee 1

*General comments: This paper summarizes a study in which the finely resolved (2.5x2.5 km) GEM-MACHPAH model was employed to investigate contributions of motor vehicles to benzene and several PAHs in ambient air across northeastern North*

[Figure]

*America. I found this paper to be a significant contribution to the field and of scientific merit, and had relatively few critical comments. I was particularly interested in the combined effects of removing benzene/PAHs at the same time as removing air quality criteria pollutants (precursors of compounds that degrade benzene/PAHs), and the differences in these effects in rural versus urban locations. The study addresses several interesting questions on a very fine spatial scale, including: -How much do vehicles contribute to ambient benzene and PAH concentrations, and how does that differ between the warm and cold seasons and across urban versus rural locations? -How do reductions in vehicle emissions play out in actual ambient concentrations? -What are the combined impacts of removing benzene/PAH oxidant precursors and benzene/PAH emissions themselves on ambient benzene/PAH concentrations? -How does particulate fractioning respond when all vehicular emissions are removed? -How do benzene/PAHs respond in general to changes in oxidant concentrations? -How sensitive is the model to doubling and halving vehicular emissions? -How does human health risk change when vehicular emissions are removed?*

(AC) Thank you for your detailed and positive review! We have addressed each of your comments below.

*Specific comments: Line 121: Can examples of "off-road mobile sources" be provided?*

(AC) Yes, we have updated the manuscript to include examples of off-road mobile sources, which include trains, boats, snowmobiles, aircraft, and others.

*Line 135: It would be helpful to have "close to unity" qualified so readers don't have to dig back through the previous Whaley paper. Plus or minus what on average?*

(AC) Thank you for the suggestion. We have quantified our statement as shown below,

Ratios of modelled to measured concentrations were generally within an order of magnitude of unity, with median values in spring-summer being lower for BENZ and PAHs with molecular weight 178-202 g mol-1 (0.31-0.86) than for PAHs with molecular weight

228-252 g mol-1 (1.8-8.4). Fall-winter values were modestly higher (1.5-9.9) though still within an order of magnitude of unity. Further details can be found in Whaley et al. (2018).

*Line 155: Lower temperatures also increase partitioning to particles for PAHs. Does the decrease in degradation rates and decrease in dilution outweigh these effects?*

(AC) Indeed, lower temperatures are generally related to increased PAH partitioning to particles. However, our statement about ambient concentrations being higher in winter than in summer refers to total (gas + particle) concentrations, and applies equally to particle-bound compounds and to those that are found exclusively or predominantly in the gas phase. Earlier studies using different particle-gas partitioning parametrizations suggest that modelled lifetimes of gaseous and particulate study compounds are similar at the regional scale (e.g., Galarneau et al., 2014, Atmos. Chem. Phys. 14:4065-4077), though we have noted that further work examining partitioning effects is warranted for future studies, particularly if larger spatial scales will be employed (line 278+).

*Line 180-ish: It was mentioned in the intro that the adoption of ZEVs was encouraged in a variety of jurisdictions. Are there differences in this "encouragement" between US and Canadian cities that could also account for the higher contributions from on-road vehicles in US cities?*

(AC) Transportation policies in Canada and the US are generally closely aligned, but our model results are based on simulations using emissions and meteorology from 2009, and market penetration of ZEVs was very low at that time, thus we don't expect this to be the cause of differences between Canada and the U.S. in our simulations for this paper. In the years since, both Canada and the US have increased their market share of EVs – for example from 2012 to 2016, the market share of EVs in both countries increased about 0.45% (from 0.15% to 0.59% in Canada, and from 0.44% to 0.91% in the US[1]). Note that these numbers include plug-in hybrid vehicles as well, not

just ZEV.

[1]https://lop.parl.ca/staticfiles/PublicWebsite/Home/ResearchPublications/BackgroundPapers/PDF/2017-27-e.pdf

*Figure S2. Is there a particular reason why the BENZ and BaP fall-winter plots are not shown? It would be handy to be able to compare seasonally for all four species chosen, or have an explanation as to why they are not shown. It is mentioned in the conclusions that future work aims to improve model representation of wintertime benzene and BaP, but (unless I missed it) I couldn't find a discussion about why their representation wasn't acceptable.*

(AC) Fall-winter BENZ and BaP were not shown for separate reasons. For BENZ, estimates of releases from residential wood combustion had not been included in the model-ready emissions, and their absence reduced our confidence in reporting motor vehicle contributions (see page 5 of the supplement) – we have added a clearer reference to this in the revised main manuscript in the Model evaluation summary. Modelled fall-winter BaP concentrations were biased high relative to available measurements, similarly reducing our confidence in the BaP fall-winter predictions, and we did not discuss vehicle contributions as a result (see lines 136-139 in original manuscript, and more detailed discussion in Whaley et al., 2018, Geosci. Model Dev. 11:2609-2632).

*Lines 189-192: Would be helpful to quantify, based on model results, how much lower concentrations would need to go to reach these criteria, and whether the complete removal of on-road vehicle sources reverses exceedances, or if additional sources would need to be removed (hard to tease out from the plots).*

(AC) Thank you for the suggestion. We agree that our model could be used as a tool to examine options for reducing airborne concentrations to levels below health-based guidelines. Unfortunately, the simulations described in this study do not provide the necessary information to do so. Guidelines tend to be promulgated at daily or annual time scales; the two that we have cited for our target jurisdiction (Toronto, Ontario,
Canada) are for BENZ and BaP on an annual basis. However, our simulations covered two seasonal periods, and BENZ and BaP results were simulated with confidence only in spring-summer. Given that the latter period does not represent typical conditions at the annual scale, we feel estimating the necessary reductions based on our results to date would not be justifiable. For example, doing so may underestimate necessary reductions given that spring-summer concentrations are lower than those at other times of the year. We hope that future applications of the model (with recommended improvements) will be useful for the analyses that you have suggested, and we appreciate your confidence in the relevance of our work.

*Line 235: This is probably discussed in previous papers, but would be handy to have a clarification here: BaP is the ONLY PAH to react with O3 on particles, even though partitioning to particles takes place for the other PAHs as well? Given the changes in O3 in rural and urban areas and the impact on BaP in the "no mobile" case, can the authors comment on how including O3 particle-phase oxidation for the other PAHs might impact their reductions? Would it be similar? How would the combination of changing OH and O3 concentrations play out for semi-volatile species? If it's easy to test this, I suggest adding it to the study.*

(AC) Yes, we have included reactions with ozone for only one particulate PAH (BaP). The literature has historically demonstrated that only BaP is substantially degraded by on-particle reactions with ozone, and robust reaction rates have not been published for other PAHs. The other particle-bound PAHs in our study are semi-volatile, meaning that only a fraction is particle-bound, and they are thought to be relatively unreactive with ozone. As a result, on-particle ozone reactions for those compounds is expected to result in negligible changes to total concentrations, and we have therefore excluded such reactions. This is clarified in the revised manuscript.

*Sensitivity to oxidant experiments shown in the Supporting Info: This is a very interesting section of the paper and it's too bad it can't be highlighted in the main text. I think it would be helpful if the authors put a section in the methods that describe how the*

*sensitivity analyses were conducted for making the S5 and S6 plots.*

(AC) Thank you for that suggestion. We have added the following text at the end of the Methods section:

The sensitivity of model results for partitioning and other parameters (e.g., oxidant concentrations) is examined in Section 3.7 and in the Supplement. Good overall performance of GEM-MACH-PAH has been demonstrated by comparison to measurements as discussed above (see Whaley et al., 2018 for further detail), and our sensitivity analyses further support the model's validity for calculating ambient concentrations and for assessing source contributions at its evaluated resolution (2.5 km grid size and seasonal time scale).

*Technical Corrections: The placement of the parentheses in the very first line of the Introduction (lines 29-31) makes the sentence difficult to read. I think it would help to have another descriptor after "air quality", like "degradation", or "problems".*

(AC) Thank you. The sentence has been re-written as:

Emissions from motor vehicles have been linked to air quality degradation (e.g., WHO(2005); Han and Naeher(2006); Zhang and Batterman(2013); Farrell et al.(2016); Zimmerman et al.(2016); Gentner et al.(2017); Wang et al.(2017)) and greenhouse gas pollution (e.g., Sims et al.(2014); Zimmerman et al.(2016); Boulton(2016); US EPA(2002)).

*Line 124: Again, parentheses are oddly placed. I suspect automated citation management software was used, so suggest combing back through the paper and removing those interrupting parentheses.*

(AC) Thank you. The sentence has been re-written as: Detailed model descriptions and evaluations of GEM-MACH have been published for pollutants other than benzene and PAHs (Moran et al.,2010,2013;Makar et al.,2015b,a;Gong et al.,2015;Whaley et al.,2018a).

*Line 165: Should be "show" instead of "shows"*

(AC) Thank you. We have corrected that.

*Line 183: Put a "to" between "concentrations" and "increase".*

(AC) Thank you, we have added that.

*Line 339: Should it read "91% PAH" instead of "91% PHEN"?*

(AC) Yes, thank you. Both are correct, but 91% PAH is what was intended as a summary of the PAH results.

\*\*\*\*\*

Anonymous Referee 2

*The introduction needs to be much clearer on why this reserach was undertaken. Additionally, the assertions need to be backed up with appropriate citations. In particular, Lines 33-35 do not seem true, so need citations to support them.*

(AC) Thank you for your review. The need for high resolution, city-scale modelling of PAHs is clear because of previous research we cite in the paper (lines 63-66 in original) showing that human exposure to PAHs is concentrated in the cities and highways. Those were observational studies, but no modelling study has been done on this subject to date. The model has the ability to quantify this source contribution – whereas measurements alone cannot.

The text in lines 33-35 contained three references in the original manuscript: US EPA, 2012; Reid and Aherne, 2006; and Government of Canada, 2018. We are uncertain which part of that text you have found to be lacking credibility; that transportation policies between Canada and the U.S. are closely aligned, or that "Government and automobile manufacturers have both pledged to further reduce emissions".

Regarding the former, we have added another reference to the revised manuscript. The

ECCC page (link [1] below) discusses the strategy to align with U.S. policies: "There is a long history of collaboration between Environment Canada (EC) and the U.S. Environmental Protection Agency (EPA) to reduce transportation emissions, largely fostered by the framework of Canada-U.S. Air Quality Agreement (AQA)."[1] There is also further evidence from a new ECCC report[2] to support "transportation policies being closely aligned in Canada and the United States", where the market share of electric vehicles in both countries has increased by a similar amount ( 0.45%) from 2012 to 2016.

[1]     https://www.canada.ca/en/environment-climate-change/corporate/international-affairs/partnerships-countries-regions/north-america/canada-united-states-vehicle-engine-emissions.html

[2] https://lop.parl.ca/staticfiles/PublicWebsite/Home/ResearchPublications/BackgroundPapers/PDF/2017-27-e.pdf

Regarding the latter, we have modified the sentence to the following in our revision: "The Canadian and US governments promote the benefits of zero emission vehicles (ZEV)[3,4], and several jurisdictions in both countries have adopted strategies to increase ZEV use (e.g., [5,6])." Indeed, the Canadian government has "established light-duty zero-emission vehicles policy sales targets of 10% by 2025, 30% by 2030, and 100% by 2040"[7].

[3]https://www.tc.gc.ca/en/services/road/innovative-technologies/zero-emission-vehicles.html    [4]https://www.energy.gov/eere/electricvehicles/electric-vehicle-benefits [5]http://www.environnement.gouv.qc.ca/changementsclimatiques/vze/index-en.htm [6]https://www.c2es.org/document/us-state-clean-vehicle-policies-and-incentives/ [7]https://www.canada.ca/en/services/environment/weather/climatechange/climate-plan/reduce-emissions.html

We regret that the motivation for the study is not clear to you. The motivation can be summarized by the following research question:

[Figure]

What is the contribution of vehicle emissions to ambient air PAH BENZ concentrations at the local (human exposure) scale?

We base our research question on the following known factors:
→ Vehicle emissions are associated with air pollution
→ Vehicle emissions represent a small fraction of total PAH BENZ emissions in national inventories

However, the system is more complex, showing the need for advanced photochemical model capable of resolving emissions and concentrations at an urban scale:
→ Due to a number of physico-chemical processes, PAH BENZ emissions are transformed between point of release and ambient air
→ Mobile PAH emissions, while small on a national basis, are emitted in a spatially heterogeneous manner (e.g., in cities and on major roadways), and the scale that is relevant to human exposure is therefore local rather than national.

We have walked through these points in our Introduction in order to set out the motivation for the work. The other reviewer for the manuscript, who found it to "be a significant contribution to the field and of scientific merit" has not indicated any such concerns, nor have colleagues who reviewed the manuscript in advance of its submission.

*I cannot comment on how this model performs relative to its peers. However, I don't see the utility in turning various block-level emission estimates and county level emissions estimates into a geographical model. There does not appear to be any need of a 2.5km resolution for the type of analyses presented here.*

(AC) Regarding your statements that you "cannot comment on how this model performs relative to its peers" and "I don't see the utility in turning...emissions...into a geographic model.": Three-dimensional chemical transport models, wherein a complete description of the processes that result in changes in concentration have been incorporated into model code, have a history in the air-quality modelling community stretching back to the late 1970's. 'Turning emissions into a geographical model' is

an oversimplification of that entire field of research. The non-linearity of the connection between emissions and ambient concentrations is well-known, and has resulted largely because of the development and application of chemical transport models over the last 40 years. These models have been used for both research and policy purposes, helping to determine the relationship between pollution sources and receptors for important environmental issues such as acid deposition and smog pollution. The additional innovations in our work were to add benzene and PAH emissions and atmospheric processing to such a model (Galarneau et al., 2014; Whaley et al 2018b); and here, to use the model to determine the relative impact of a specific source sector on PAH concentrations, hence towards human exposure in populated areas.

Understanding the contribution of a potentially important pollution source such as vehicle exhaust to ambient air, where human exposure occurs, is the crux of the "need of a 2.5km resolution". Elevated pollutant concentrations tend to be found in urban areas where fine-scale modelling is required to properly represent spatial gradients. Measurements in ambient air, which are expensive for PAHs relative to other common air pollutants, are not available at such a fine scale over entire regions. Furthermore, determining source contributions from ambient air measurements is problematic, given the overlap in source signatures and differences in atmospheric processing that occur for these compounds. Hence, chemical transport models at fine spatial scale are tools of unparalleled utility for examining the importance of air pollution sources for the air to which humans are exposed.

*In examining the referenced article Whaley et al 2018b, it is not clear that the GEMMACH-PAH model has the accuracy and precision to really describe the difference the authors say it is between the two scenarios. The variability across sites and seasons seems that it is larger than the observed difference. For example, the percent reduction of the PF of FLRT in winter looks to be 20% in Figure 6f. In Whaley et al 2018b, the model to measurement ratio for FLRT varies across sites from -10 to 10. It seems that this degree of uncertainty makes it hard to believe the model is able to tell*

*the difference between a change of 20% and a change of 200%.*

(AC) We agree that adequate performance from an atmospheric chemical transport model such as GEM-MACH-PAH must be demonstrated before it is used to answer questions about source contributions. At the same time, a predicted spatial variation in a simulated field such as the model to measurement ratio for FLRT should not be confused with the impact of an emissions scenario which results in a reduction in concentrations across the entire domain. The model-to-measurement ratio for the base case mentioned is a prediction of spatial variability in error, not a measure of uncertainty in the model predictions across the domain. For domain-wide impact predictions such as the percent reduction of PF, the model's bias with respect to observations is the relevant metric of performance. As presented in the 2018 paper, and reiterated in the current manuscript, simulated seasonal concentrations were found to be unbiased relative to measurements for all reported compounds. This finding was based on measurements from locations associated with a variety of land uses (e.g., urban, industrial, rural, etc.) and concentration levels, and represents the most extensive assessment of model performance ever conducted for PAHs. We also note that the reviewer's cited rage is based on daily values (Figure 6b in Whaley et al., 2018b) whereas the vehicle contributions in the current manuscript are based on seasonal averages, which exhibit less inter-site variability.

Though atmospheric chemical transport models are not perfect, validated models such as ours are nevertheless useful for assessing source contributions. We used our validated model to examine vehicle contributions by taking the difference between two simulations ("base" with all emissions on, and "no mobile" with vehicle emissions off). Given that process representation and non-vehicle emission uncertainties will be the same in both simulations, the impact of those uncertainties are substantially reduced when estimating vehicle contributions. This is also the strandard "scenario emissions simulation" practice in use by the air-quality modelling community in providing advice on the impact of emissions over the last 40 years. As a result, we have high confidence

in the resulting vehicle contributions that we have reported.

As an additional measure, we note that in our submitted manuscript we examined potential uncertainties in vehicle emissions, which do not cancel out in our difference scenario. Our tests showed that changes in simulated concentrations were sensitive to, but consistent with, the emission perturbations (see lines 287-296 of the original manuscript and Section V of the Supplement), lending further confidence to our mobile source contribution estimates.

*I'm also not completely clear on what the difference is between the Whaley 2018 model and the model used in this paper.*

(AC) The model used in both papers is the same, and is described in Section 2.1 of the manuscript. The Whaley 2018b paper provided the technical background of the model development and evaluation, and the current paper describes the application of the model to answer the research question noted above (viz., what is the contribution of vehicle emissions to ambient air concentrations of PAHs at the local (human exposure) scale?).
* * *